# Deep Fractional Fourier Transform

**Hu Yu**[*,†,1]    **Jie Huang**[*,1]    **Lingzhi Li**[2]    **Man Zhou**[1]    **Feng Zhao** [‡,1,]
[1]University of Science and Technology of China    [2]Alibaba Group
{yuhu520, hj0117, manman}@mail.ustc.edu.cn  fzhao956@ustc.edu.cn

## Abstract

Existing deep learning-based computer vision methods usually operate in the spatial and frequency domains, which are two orthogonal **individual** perspectives for image processing. In this paper, we introduce a new spatial-frequency analysis tool, Fractional Fourier Transform (FRFT), to provide comprehensive **unified** spatial-frequency perspectives. The FRFT is a unified continuous spatial-frequency transform that simultaneously reflects an image's spatial and frequency representations, making it optimal for processing non-stationary image signals. We explore the properties of the FRFT for image processing and present a fast implementation of the 2D FRFT, which facilitates its widespread use. Based on these explorations, we introduce a simple yet effective operator, Multi-order FRactional Fourier Convolution (MFRFC), which exhibits the remarkable merits of processing images from more perspectives in the spatial-frequency plane. Our proposed MFRFC is a general and basic operator that can be easily integrated into various tasks for performance improvement. We experimentally evaluate the MFRFC on various computer vision tasks, including object detection, image classification, guided super-resolution, denoising, dehazing, deraining, and low-light enhancement. Our proposed MFRFC consistently outperforms baseline methods by significant margins across all tasks. Our code is released publicly at `https://github.com/yuhuUSTC/FRFT`.

## 1   Introduction

Vanilla convolution in the spatial domain has been the dominant approach in Convolutional Neural Networks (CNNs) due to its advantages of fewer parameters with parameter sharing and translation invariance. However, CNNs suffer from a limited receptive field, especially in early layers. To address this issue, several methods have attempted to convert signals from the spatial domain into the frequency domain using the Fourier Transform (FT), which offers a global receptive field according to the spectral convolution theorem [1] and is physically meaningful. However, FT has limitations due to its reliance on stationary signals and lack of spatial-frequency properties. As a global transformation, it captures the overall spectrum of an image but cannot express the local spatial characteristics of the image. This makes FT optimal for stationary signals but not for non-stationary ones, such as images. Additionally, the spatial and frequency domains are only two special and orthogonal perspectives of signals. **Then, we wonder if there is an optimal tool for processing non-stationary image signals and providing an intermediate and unified perspective of an image.**

To address the aforementioned question, we introduce a unified spatial-frequency [1] analysis tool called the Fractional Fourier Transform (FRFT), which offers several advantages. First, the basis function of the FRFT is the linear frequency modulation function, making it well-suited for processing non-stationary image signals. As shown in Figure 1(a), non-stationary image signals are not separable

---

[*]Equal Contribution.      [†]Work partially performed during internship at Alibaba DAMO Academy.
[‡]Corresponding Author.

    [1]Also, a time-frequency analysis tool for time-varying signals such as video, speech, and radar signals.

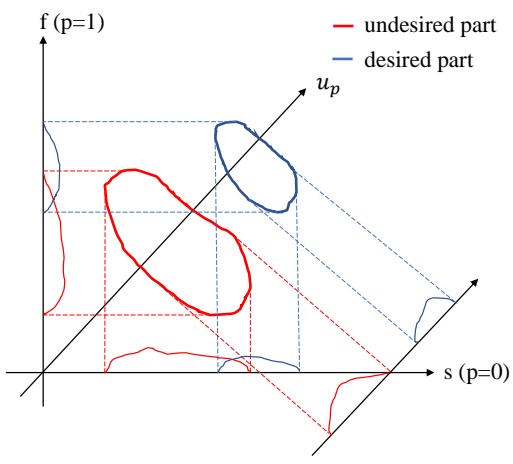
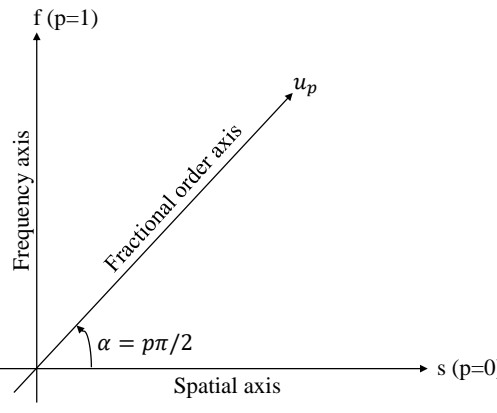

(a) FRFT is inherently optimal for non-stationary image signals and is capable of separating them effectively along the fractional order axis..

(b) FRFT is a generalized version of Fourier transform, providing a unified and intermediate perspective for image processing.

Figure 1: Merits of FRFT for image processing. The relationship between FRFT and spatial-frequency plane rotation is detailed in subsection 3.

when projected onto the spatial or frequency axis, but they are separable on the fractional order axis, enabling the FRFT to extract the desired features from an image. Secondly, as illustrated in Figure 1(b), the FRFT generalizes the traditional Fourier Transform (FT) to the entire spatial-frequency domain with an additional parameter known as the order $p$, which is related to a signal rotation angle in the spatial-frequency plane. In spatial-frequency analysis, a signal is represented in a plane where the spatial and frequency domains are orthogonal axes, as depicted in Figure 1. For a signal $f(x)$ along the spatial axis, its traditional FT $F(w)$ corresponds to the frequency axis, meaning a 90° rotation in the spatial-frequency plane. The FRFT, on the other hand, can be interpreted as the $p$th-order counterclockwise rotation (rotation angle $\alpha = p\pi/2$) in the spatial-frequency plane [2], where the spatial domain denotes order 0 and the frequency domain corresponds to order 1. The axis $u_p$ in Figure 1 represents a rotated axis, also known as the fractional order axis. Therefore, while the spatial and frequency domains are only two special orthogonal orders, the FRFT of fractional order $p$ can provide any intermediate representation between the spatial and frequency domains. This property allows for a unified spatial-frequency characterization and provides multiple perspectives for image signal processing.

Despite its promising merits, the FRFT faces a challenge in terms of fast implementation. Unlike the Fast Fourier Transform (FFT) for the FT, there is currently no encapsulated package available for the fast implementation of the FRFT. To address this, we develop a fast implementation of the 2D FRFT. Building on these advancements, we have introduced a simple yet effective operator, the Multi-order Fractional Fourier Convolution (MFRFC), which processes images from multiple perspectives in the spatial-frequency plane. The MFRFC consists of three different order paths: a spatial perspective, a frequency perspective, and a unified spatial-frequency representation perspective. These various order operations are applied to disjoint subsets of feature channels, and the updated feature maps are aggregated to produce the output. MFRFC inherently preserves the advantages of both spatial and frequency operations. More importantly, the additional unified spatial-frequency representation reflect the spatial-frequency joint characteristics of signals. Our proposed MFRFC is a general and basic operator that can be easily integrated into existing methods. We have experimentally evaluated our operators on various tasks, including object detection, image classification, guided super-resolution, denoising, dehazing, deraining, and low-light enhancement. The results consistently demonstrate significant performance improvements across all the evaluated tasks.

## 2    Related Work

**Spectral neural networks.**    In recent years, there has been a surge of interest in spectral neural network research [3, 4, 5, 6, 7, 8, 9]. Previously used to exclusively speed up convolutions, the spectral

domain now also serves as a critical building element for deep networks. For example, the authors in [3] proposed fast Fourier convolution, which has the main hallmarks of non-local receptive fields and cross-scale fusion within the convolutional unit. Rippel et al. [4] developed the spectral pooling, which performs dimensionality reduction by truncating the representation in the frequency domain. A frequency-domain dynamic pruning scheme was designed in [6] to exploit the spatial correlations for network compression and acceleration. Some methods [10, 11, 12, 13, 14, 15] also employ Fourier analysis into image restoration tasks. Our method contributes to the above-mentioned research by generalizing existing methodologies and constructing a unified spatial-frequency operating unit that processes various perspectives simultaneously.

**Spatial-frequency analysis tool.** Spatial-frequency analysis is a powerful method that provides joint distribution information in both the spatial and frequency domains, offering a clear description of the relationship between signal frequency and spatial variation. There are several common spatial-frequency analysis tools, including the short-time Fourier transform (STFT) [16, 17] and wavelet transform (WT) [18]. The STFT applies a local spatial window to the signal and performs the Fourier transform within this window, assuming that the signal within the time window is stationary. However, it faces the dilemma of frequency and spatial resolution balance. The WT tackles this limitation by adopting an adaptive frequency window. However, the WT faces its limitations, including the optimal choice of wavelet basis function, higher computational complexity, and signal distortion. In this paper, we introduce another Spatial-frequency analysis tool, FRFT. More importantly, we achieve the fast implementation of 2D FRFT and apply it to different image processing tasks.

**Fractional Fourier Transform.** The FRFT is a powerful spatial-frequency analysis tool used for signal processing due to its unique capability of capturing the non-stationary characteristics of signals [19]. For example, Ozaktas et al. [20] demonstrated that the Wigner-Ville Distribution of the pth order Fractional Fourier Transform is obtained from the $p\pi/2$ angle rotation of the original signal's WVD. The FRFT has also found applications in multimedia, including digital watermarking, image compression, and image encryption systems [21, 22, 23]. In communication, the FRFT has been implemented in wireless communications and transmission schemes [24, 25]. Additionally, Zhang et al. [26] investigated the use of FRFT as a framework for biomedical signal detection. In this work, we explore the property of FRFT for deep learning. Based on our exploration, we propose a novel convolutional operator that takes advantage of the unique properties of FRFT.

## 3   Preliminary of FRFT for deep learning

The Fractional Fourier Transform (FRFT) is a powerful and versatile analysis tool that has been relatively under-explored in the context of deep learning. In this section, we aim to reveal the unique properties of FRFT and explain why it is a valuable addition to the deep learning toolkit.

**Definition**   We first introduce the mathematical definition of FRFT. The 1-D p-th order FRFT of a signal $x(u_0)$, where $u_0$ represents the initial domain, i.e., time, spatial, frequency, or even an intermediate domain, is defined as follows:

$$F^p\{x(u_0)\} = X(u_p) = \int_{-\infty}^{\infty} K_p(u_0, u_p)\, x(u_0)\, du_0. \tag{1}$$

Specifically, the kernel $K_p(u_0, u_p)$ is given by the following expression:

$$K_p(u_0, u_p) = \begin{cases} A_\alpha e^{j\left(u_0^2 \cot\alpha/2 - u_p u_0 \csc\alpha + u_p^2 \cot\alpha/2\right)}, & \alpha \neq n\pi \\ \delta(t - u), & \alpha = 2n\pi \\ \delta(t + u), & \alpha = (2n \pm 1)\pi \end{cases} \tag{2}$$

where $A_\alpha = \sqrt{1 - j\cot\alpha}$, and $\alpha = \frac{p\pi}{2}$. Eq. 1 can be called either a p-th order Fractional Fourier Transform of signal $x(u_0)$ or a Fractional Fourier Transform of signal $x(u_0)$ at angle $\alpha$. For FRFT, the basis function is linear frequency modulation (LFM) signal, as shown in the kernel function, instead of the sine function basis in Fourier transform. Besides, when $\alpha = \pi/2$, the formula of FRFT degrades to the formula of FT, which is proved with details in the supplementary material.

**Fast discrete implementation of FRFT**   FT has a well-established fast digital computation algorithm that can compute it in N log N time [27]. However, despite the availability of this algorithm,

there is currently no encapsulated package for the fast implementation of FRFT. Furthermore, most existing implementation methods cannot match the computational speed of Fourier transform in practical applications. To address this issue, we have adopted a matrix multiplication implementation [28] that was originally developed for natural language processing and adapted it for the field of computer vision. Through this adaptation, we have developed an improved fast version of FRFT that is as fast as Fourier transform in practical settings. This implementation offers a valuable tool for researchers and practitioners seeking to leverage the benefits of FRFT in their work.

**FRFT, Wigner distribution, and spatial-frequency plane rotation**    The Wigner distribution of signal $f(x)$ is defined as follows:

$$W_f(x,\nu) = \int_{-\infty}^{\infty} f\left(x + x'/2\right) f^*\left(x - x'/2\right) e^{-i2\pi\nu x'} dx'. \tag{3}$$

The function $W_f(x,\nu)$ provides a distribution of signal energy over time and frequency, and it serves as a key link between FRFT and spatial-frequency plane rotation. Specifically, the Wigner distribution of the p-th order FRFT of a signal $f(x)$ is equivalent to a counterclockwise rotation of $W_f(x,\nu)$ by an angle of $\alpha = p\pi/2$ degrees [27].

**Non-stationary signal**    A non-stationary signal is one whose statistical properties vary with time or space, including the mean, variance, and frequency. In contrast to stationary signals, which are often artificially created, natural signals are almost always non-stationary. Natural images are also non-stationary signals since their statistical properties change significantly over space, depending on the content of the image. For example, the mean and variance of an image may vary depending on factors such as lighting conditions, color palette, or image composition. Additionally, images may contain non-stationary features such as edges, textures, and patterns that change over space. These non-stationary features play a crucial role in many computer vision tasks. Thus, FRFT is well-suited for the processing of the non-stationary image signal.

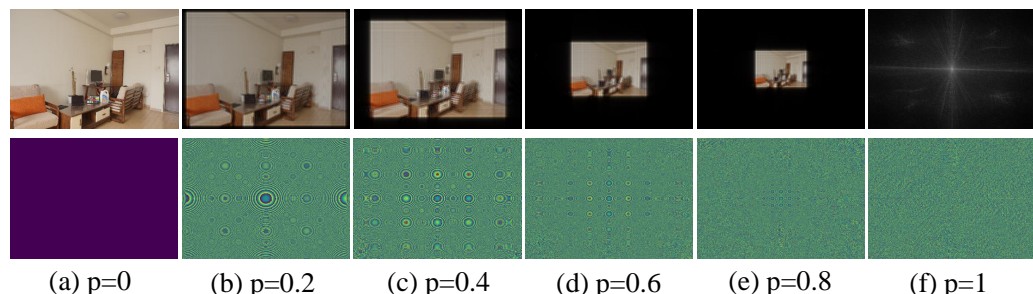

| (a) p=0 | (b) p=0.2 | (c) p=0.4 | (d) p=0.6 | (e) p=0.8 | (f) p=1 |

Figure 2: The amplitude and phase spectrums of the FRFT of different orders. The upper row is amplitude spectrums, and the bottom is phase spectrums. Note that the fractional order's amplitude spectrum contains the signals' spatial-frequency joint characteristics.

**The amplitude and phase spectrums of image in FRFT**    As shown in Fig. 2, the first row is the amplitude spectrum of FRFT in different orders, and the second is the phase spectrum. For the amplitude spectrum, when the order is small, the outline and detail information of the image can be clearly observed. With the increase of the order, the image gradually becomes blurred, and the energy becomes more and more concentrated. It shows that the spatial domain information in the amplitude spectrum decreases with the increase of the transformation order. In contrast, the frequency domain information increases with the rise of the transformation order.

**The comparison between spatial image, FT and FRFT**    Table 1 presents a comparison between spatial image, Fourier Transform (FT), and Fractional Fourier Transform (FRFT). Traditional FT is a powerful tool for processing stationary signals, with the sine function as the basis function. In contrast, FRFT is a spatial-frequency analysis tool that is particularly effective for processing non-stationary signals, with the linear frequency modulation signal as the basis function. Spatial and FT are two orthogonal individual perspectives, with the spatial image corresponding to the horizontal s-axis in the spatial-frequency plane shown in Fig. 1, and FT corresponding to the vertical f-axis in the same plane. On the other hand, FRFT can represent a unified spatial-frequency perspective in the

same plane, providing a more comprehensive and integrated view of the signal. Additionally, FRFT introduces the concept of fractional transformation order, which allows for the analysis of the signal's spatial and frequency domain information. In contrast, the FT of a signal only contains frequency domain information, limiting its scope of analysis.

Table 1: Comprehensive comparison between original spatial image, Fourier transform, and FRFT.

| Difference | Spatial | Fourier transform | FRFT |
|---|---|---|---|
| Applicable signal | * | Stationary | Time varying non-stationary |
| Basis function | * | Sine | Linear frequency modulation |
| Rotation angle | 0° | 90° | $\alpha$ (any) |
| Generality | individual | individual | unified |
| Information | Spatial | Frequency | Spatial & Frequency |

## 4 FRFT operator design

Based on the property of FRFT in image processing and our fast implementation of FRFT, we devise a simple yet effective operator, the Multi-order Fractional Fourier Convolution (MFRFC).

### 4.1 MFRFC

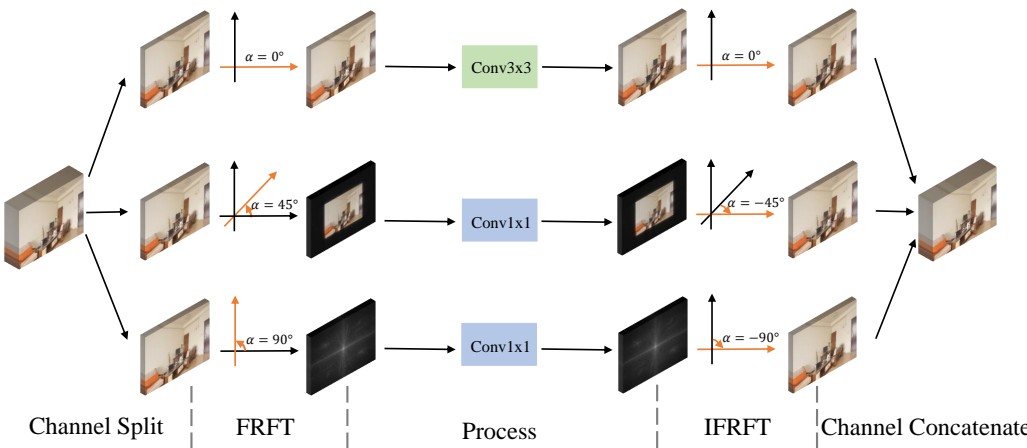

Figure 3: Architecture design of the Multi-order Fractional Fourier Convolution. MFRFC is comprised of three different order paths: a spatial (p=0) path, a frequency (p=1) path, and a unified spatial-frequency (p=0.5) path. IFRFT denotes Inverse FRactional Fourier Transform.

The architecture of our proposed Multi-order Fractional Fourier Convolution (MFRFC) is illustrated in Fig. 3. MFRFC comprises five critical operations: a channel split that equally divides the input features into three parts along the channel dimension, a 2D discrete Fractional Fourier Transform that converts the input features to the fractional domain with three different orders, separate processing of these three fractional domains, a 2D inverse Fractional Fourier Transform that converts the fractional domain features back to the spatial domain, and a channel merge that concatenates the three features along the channel dimension. These five operations work in concert to create a powerful and flexible convolutional layer that can leverage the unique properties of FRFT to analyze non-stationary signals and improve the performance of deep learning models.

The key component of MFRFC is the three different order paths: a spatial (p=0) path that conducts ordinary convolutions, a spectral (p=1) path that operates in the spectral domain, and a fractional order (p=0.5) path that operates in the unified spatial-frequency domain. After the channel concatenation operation, we add a full channel convolution to fuse the information from different perspectives.

## 4.2 Merits of MFRFC

The design of MFRFC offers several advantages over vanilla convolution and Fourier transform. Firstly, it preserves all the key merits of vanilla convolution and Fourier transform, such as parameter sharing and global receptive field. For instance, spectral theory demonstrates the existence of operator duality between convolution in the spatial domain and element-wise multiplication in the spectral domain, as shown in the following equation:

$$F(f(x_1) * f(x_2)) = F(x_1) \odot F(x_2),    \tag{4}$$

where $*$ and $\odot$ denote convolution and element-wise product, respectively. This enables FT with image-size receptive field.

Moreover, MFRFC provides the added advantage of analyzing features from multiple perspectives. In particular, the introduction of the fractional domain complements the missing perspective that is optimal for non-stationary signal analysis. The fractional domain is inherently well-suited for separating and extracting non-stationary features, offering a powerful tool for analyzing image signals with varying statistical properties over space.

## 4.3 Fractional order selection

Regarding the path number, we select three paths: traditional spatial and frequency paths, as well as an additional fractional path. This approach allows us to leverage the two orthogonal axes of the spatial-frequency plane, as well as the intermediate fractional axis, to provide more comprehensive and representative signal projection perspectives.

In terms of fractional order selection, our experimental results demonstrate that different orders can improve performance [2]. For simplicity, we choose a fixed order p=0.5 based on empirical observations. While this choice is simple, it still provides a noticeable performance improvement on various tasks, demonstrating the effectiveness of incorporating FRFT, which is a key focus of our work. In addition, we also provide an extended version of MFRFC, called MFRFC++, in the supplementary material, which enables the network to learn the optimal order adaptively.

## 4.4 Complexity analysis

Table 2 compares two major complexity metrics of the Multi-order Fractional Fourier Convolution (MFRFC) and vanilla convolution. Thanks to the fast implementation of FRFT, the complexity of FRFT and inverse FRFT is omitted since they are parameter-free and their time complexities are negligible compared to other computation costs. As shown, the computational cost of FRFT is comparable to that of vanilla convolution. Therefore, in specific usage, we can simply insert the MFRFC once into different networks to improve performance with negligible parameter increase.

Table 2: Parameter counts and FLOPs for vanilla convolution, separate components in MFRFC, and entire MFRFC, respectively. $C_1$ and $C_2$ are the numbers of input and output channels, respectively. We equally split the input feature channel for every path, but for order=0.5 and order=1 paths, the concatenation of the real and imaginary parts doubles the channel number. So the final channel number for these two paths is $\frac{2}{3}$ of the original. $H$ and $W$ collectively define spatial resolution. $K$ is the convolutional kernel size. For clarity, here, stride and padding are not considered.

|  | Params | FLOPs |
|---|---|---|
| vanilla | $C_1 C_2 K^2$ | $C_1 C_2 K^2 H W$ |
| Path1 (order=0) | $\frac{1}{9} C_1 C_2 K^2$ | $\frac{1}{9} C_1 C_2 K^2 H W$ |
| Path2 (order=0.5) | $\frac{4}{9} C_1 C_2$ | $\frac{4}{9} C_1 C_2 K^2 H W$ |
| Path3 (order=1) | $\frac{4}{9} C_1 C_2$ | $\frac{4}{9} C_1 C_2 K^2 H W$ |
| Path fusion | $C_1 C_2 K^2$ | $C_1 C_2 K^2 H W$ |
| MFRFC | $(1 + \frac{1}{9} + \frac{8}{9K^2}) C_1 C_2 K^2$ | $(1 + \frac{1}{9} + \frac{8}{9K^2}) C_1 C_2 K^2 H W$ |

---

[2]The relationship between order and performance is provided in the supplementary material.

# 5 Experiments

To demonstrate the effectiveness of our proposed convolutional operator MFRFC, we conduct extensive experiments over multiple computer vision tasks, including object detection, image classification, guided super-resolution, denoising, dehazing, deraining, and low-light enhancement.

## 5.1 Evaluation Protocols

**Object detection.** Following [29], the PASCAL VOC 2007 and 2012 training sets [30] are adopted for training, and the PASCAL VOC 2007 testing set is used for evaluation. We employ the FPN-based Faster RCNN with ResNet50 backbone as baseline.

**Image classification.** We verify our method on the image classification task with the CIFAR-10 dataset [31]. For the baseline, we employ the classical classification method: ResNet [32].

**Image denoising.** Following [33], to evaluate our method on the image de-noising task, we employ the widely-used SIDD dataset as training benchmark. Further, the corresponding performance evaluation is conducted on the remaining validation samples from the SIDD dataset [34]. For the baseline, we employ the representative denoising method: DnCNN [35].

**Image enhancement.** We verify our method on the image enhancement task with the commonly used dataset, LOL [36]. Further, we adopt the two classical baselines, SID [37] and DRBN [38].

**Image de-hazing.** Following [39], to evaluate our method on the image dehazing task, we employ ITS and SOTS indoor [40] as our training and testing datasets. For the baseline, we adopt the representative dehazing methods: GridNet [41] and MSBDN [42].

**Image de-raining.** For validation, we used the widely-used standard benchmark dataset Rain200H, as described in [43]. As baselines, we use the representative de-raining methods: PReNet [43].

**Guided image super-resolution.** Following [44, 45], we adopt pan-sharpening, the representative task of guided image super-resolution for evaluating the effectiveness of the MFRFC. The WorldView II, WorldView III, and GaoFen2 in [44, 45] are used for evaluation. We employ two different network designs for validation: PanNet [46], and FusionNet [47]. Several widely used image quality assessment (IQA) metrics are employed to evaluate the performance, including the peak signal-to-noise ratio (PSNR), the structural similarity index (SSIM), the spectral angle mapper (SAM) [48], and the relative dimensionless global error in synthesis (ERGAS) [49].

## 5.2 Implementation details

**Baseline variant.** Besides our MFRFC operator, we also validate a baseline variant called Spatial-Frequency Convolution (SFC). Specifically, SFC has two order paths, a spatial path (order=0) and a frequency path (order=0). Our MFRFC has an additional fractional path than SFC and is supposed to perform better than SFC.

**Three configurations.** We re-implement the baseline methods following the settings in the corresponding paper. In this paper, we denote baseline models as "Original", baseline models equipped with SFC as "SFC", and baseline models provided with MFRFC as "MFRFC". We present an example usage of SFC and MFRFC in the supplementary material.

## 5.3 Analysis and visualization

**Quantitative Comparison.** We compare model performance over different configurations, as described in implementation details. The quantitative results are reported in Table 3, 4, 5, 6, 7, 8, and 9. It is evident that, by integrating with our proposed MFRFC operator, all "MFRFC" obtain performance gain over the "Original" and "SFC" baselines on all the datasets in the evaluated task, which fully validates the effectiveness of our method.

**Qualitative Comparison.** Due to the page limits [3], we only show the visual results over the representative tasks of guided image super-resolution, image deraining, and image dehazing in Fig. 4, 5, and 6, respectively, to better illustrate the effectiveness of MFRFC. It can be easily figured out

---

[3]More visual comparison results are reported in the supplementary materials.

that "MFRFC" generates the highest-fidelity results that also look perceptually close to the reference ground-truths. For example, zooming in the red box region in Figure 4, the "MFRFC" model can better recover the texture details. Similar conclusions also apply to Fig. 6.

Table 3: Quantitative comparison over object detection.

| Model | Methods | AP50 | mAP |
|---|---|---|---|
| Faster RCNN | Original | 77.30 | 77.26 |
| | SFC | 77.90 | 77.89 |
| | MFRFC | 78.00 | 77.96 |

Table 4: Quantitative comparison on image classification.

| Model | Methods | Acc |
|---|---|---|
| ResNet | Original | 95.23 |
| | SFC | 95.34 |
| | MFRFC | 95.43 |

Table 5: Quantitative comparison on image denoising.

| Model | Methods | PSNR | SSIM |
|---|---|---|---|
| DnCNN | Original | 37.1992 | 0.8904 |
| | SFC | 37.3454 | 0.8866 |
| | MFRFC | 37.6084 | 0.8915 |

Table 6: Quantitative comparison on image deraining.

| Model | Methods | PSNR | SSIM |
|---|---|---|---|
| PReNet | Original | 29.041 | 0.891 |
| | SFC | 29.498 | 0.901 |
| | MFRFC | 29.735 | 0.906 |

Table 7: Comparison of quantitative results over guided image super-resolution.

| Model | Methods | WorldView-II | | | | GaoFen2 | | | |
|---|---|---|---|---|---|---|---|---|---|
| | | PSNR↑ | SSIM↑ | SAM↓ | ERGAS↓ | PSNR↑ | SSIM↑ | SAM↓ | EGAS↓ |
| PanNet | Original | 40.817 | 0.9626 | 0.0257 | 1.0557 | 43.0659 | 0.9685 | 0.0178 | 0.8577 |
| | SFC | 41.2118 | 0.9663 | 0.0236 | 0.9901 | 44.4965 | 0.9801 | 0.0161 | 0.7761 |
| | MFRFC | 41.3428 | 0.9667 | 0.0233 | 0.9898 | 44.6245 | 0.9804 | 0.0146 | 0.6551 |
| FusionNet | Original | 41.0708 | 0.9651 | 0.0248 | 0.9978 | 44.8852 | 0.9810 | 0.0140 | 0.6661 |
| | SFC | 41.3941 | 0.9674 | 0.0232 | 0.9898 | 45.1071 | 0.9810 | 0.0138 | 0.6413 |
| | MFRFC | 41.5829 | 0.9676 | 0.0229 | 0.9766 | 45.2534 | 0.9876 | 0.0137 | 0.6377 |

Table 8: Quantitative comparison over image dehazing.

| Model | Methods | PSNR | SSIM |
|---|---|---|---|
| GridNet | Original | 32.56 | 0.9847 |
| | SFC | 32.97 | 0.9856 |
| | MFRFC | 33.39 | 0.9887 |
| MSBDN | Original | 32.19 | 0.9813 |
| | SFC | 32.26 | 0.9825 |
| | MFRFC | 32.45 | 0.9838 |

Table 9: Quantitative comparison on low-light image enhancement.

| Model | Methods | PSNR | SSIM |
|---|---|---|---|
| SID | Original | 20.1062 | 0.7895 |
| | SFC | 21.9222 | 0.7988 |
| | MFRFC | 22.3666 | 0.7998 |
| DRBN | Original | 19.7931 | 0.8361 |
| | SFC | 21.8372 | 0.8356 |
| | MFRFC | 21.9916 | 0.8426 |

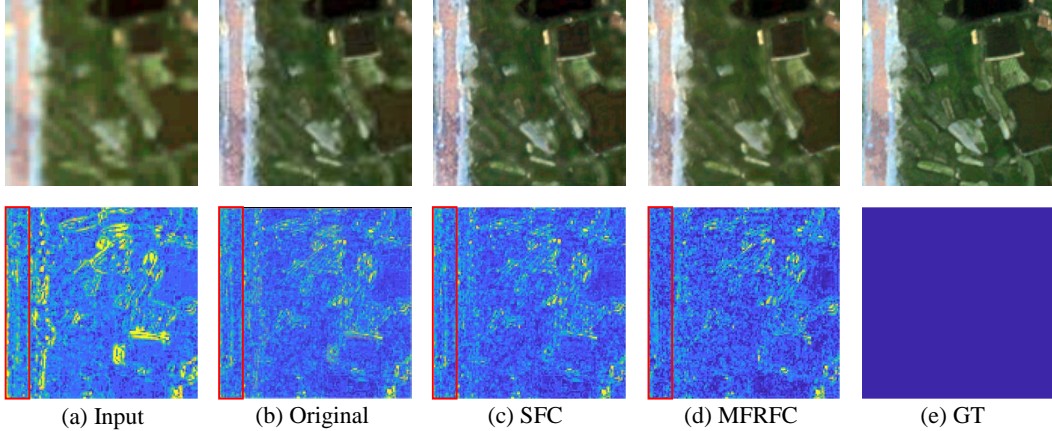

(a) Input    (b) Original    (c) SFC    (d) MFRFC    (e) GT

Figure 4: The visual comparison of PANNET over WorldView-II. The bottom row is error map.

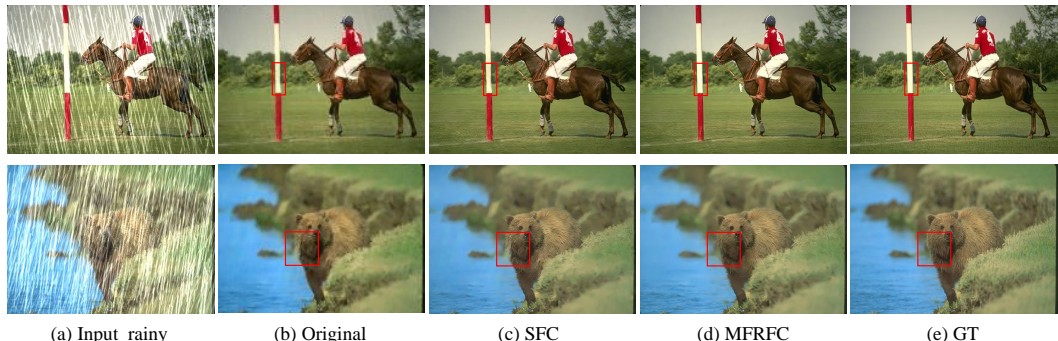

| (a) Input_rainy | (b) Original | (c) SFC | (d) MFRFC | (e) GT |

Figure 5: The visual comparison of PReNet over the Rain200H.

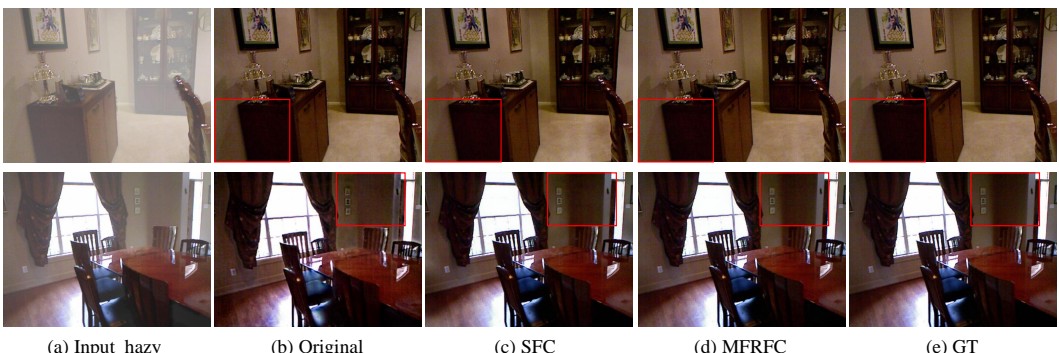

| (a) Input_hazy | (b) Original | (c) SFC | (d) MFRFC | (e) GT |

Figure 6: The visual comparison of GridNet over SOTS.

## 6  Limitations and Discussions

We evaluate the effectiveness of the proposed method over several computer vision tasks, and we will conduct more extensive experiments on other computer vision tasks (e.g., image segmentation and video processing). Furthermore, we recognize that the optimal fractional order may vary depending on the specific task and may be critical for the further understanding and application of FRFT in specific task. To emphasize, the critical contribution of this work is that we clear the way to the popularization of FRFT, and make the first attempt to integrate FRFT into a basic operator.

## 7  Conclusion

In this paper, we introduce a new spatial-frequency analysis tool, FRFT, to generalize existing FT and provide comprehensive perspectives for vision tasks. The FRFT is a unified spatial-frequency transform, simultaneously reflecting an image's spatial and frequency representations. Notably, we explore the properties of FRFT for image processing and achieve the fast implementation of 2D FRFT. Besides, we devise the convolutional operator MFRFC, which has the impressive merits of processing an image from more perspectives in the spatial-frequency plane. The MFRFC is a general convolutional operator and thus can be easily integrated into existing methods to improve their performance further. Extensive experiments have demonstrated the effectiveness of our method.

### Broader Impact

The past decades have witnessed the development of computer vision driven by convolution neural networks. However, vanilla spatial convolution is only one common perspective of signal processing. Some spatial-frequency combination methods have already exhibited the potential of two-perspective signal processing. As a general version of FT, FRFT is quite promising for image processing but is less explored largely due to the missing fast discrete implementation algorithm and unclear properties in image processing. In this paper, we have cleared all these obstacles to the popularization of

FRFT and made the first attempt to integrate FRFT into a novel convolutional operator, which has demonstrated its effectiveness in various computer vision tasks. Besides, we hold the belief that there is much more space for the further exploration of FRFT in deep learning. Additionally, our work may inspire the rethinking of neural networks from more diverse perspectives.

## Acknowledgements

This work was supported by the JKW Research Funds under Grant 20-163-14-LZ-001-004-01, and the Anhui Provincial Natural Science Foundation under Grant 2108085UD12. We acknowledge the support of GPU cluster built by MCC Lab of Information Science and Technology Institution, USTC.

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
