# Supplementary Materials for "Deep Fractional Fourier Transform"

**Hu Yu**[*,†,1]    **Jie Huang**[*,1]    **Lingzhi Li**[2]    **Man Zhou**[1]    **Feng Zhao** [‡,1],
[1]University of Science and Technology of China    [2]Alibaba Group
{yuhu520, hj0117, manman}@mail.ustc.edu.cn  fzhao956@ustc.edu.cn
https://github.com/yuhuUSTC/FRFT

This supplementary document is organized as follows:

Section 1 shows the proof that the formula of FRFT degrades to that of FT when $\alpha = \pi/2$.

Section 2 shows the discrete implementation of 2D FRFT.

Section 3 shows the relationship between fractional order and performance and the extended version MFRFC++ with adaptive order.

Section 4 shows the experimental results with single branch.

Section 5 shows the architecture design of SFC and example usage of SFC and MFRFC.

Section 6 introduces the periodicity of FRFT.

Section 7 introduces the energy distribution of FRFT.

Section 8 gives a specific example of the potential of FRFT for image restoration task.

Section 9 shows more visual results.

Section 10 provides more visualization results of feature maps.

## 1   The formula of FRFT degrades to that of FT when $\alpha = \pi/2$

In the main body, we claim that FT is a special case of the FRFT, and the formula of FRFT degrades to that of FT when $\alpha = \pi/2$. In this section, we show the detail proof. We firstly revisit the formula of FRFT mentioned in the main body. The 1D pth-order FRFT of a signal $f(u_0)$ is defined as follows:

$$F^p \{f (u_0)\} = F (u_p) = \int_{-\infty}^{\infty} K_p (u_0, u_p) f (u_0) \, du_0. \tag{1}$$

Specifically, the kernel $K_p (u_0, u_p)$ is given by the following expression:

$$K_p(u_0, u_p) = \begin{cases} A_\alpha e^{j\left(u_0^2 \cot \alpha/2 - u_p u_0 \csc \alpha + u_p^2 \cot \alpha/2\right)}, & \alpha \neq n\pi \\ \delta(t - u), & \alpha = 2n\pi \\ \delta(t + u), & \alpha = (2n \pm 1)\pi \end{cases} \tag{2}$$

where $A_\alpha = \sqrt{1 - j \cot \alpha}$, and $\alpha = \frac{p\pi}{2}$.

Then, when $\alpha = \pi/2$ , we get $p = 1$, $\cot \alpha = 0$, $\csc \alpha = 1$, and $A_\alpha = \sqrt{1 - j \cot \alpha} = 1$. Thus, the kernel $K_{p=1} (u_0, u_1)$ and FRFT are simplified as follows:

$$K_{p=1}(u_0, u_1) = e^{-j u_1 u_0}. \tag{3}$$

$$F^{p=1} \{f (u_0)\} = \int_{-\infty}^{\infty} f(u_0)e^{-j u_1 u_0} du_0. \tag{4}$$

---

*Equal Contribution.    †Work partially performed during internship at Alibaba DAMO Academy.
‡Corresponding Author.

37th Conference on Neural Information Processing Systems (NeurIPS 2023).

If $u_0$ represents the spatial domain ($u_0 = x$), then $u_1$ denotes the frequency domain ($u_1 = w$), and FRFT can be written as:

$$F^{p=1}\{f(u_0)\} = \int_{-\infty}^{\infty} f(x)e^{-jwx}dx, \tag{5}$$

which is exactly the formula of FT. We successfully and detailly proved the claim that the formula of FRFT degrades to that of FT when $\alpha = \pi/2$.

## 2 Discrete implementation of the FRFT (DFRFT)

The discrete implementation of the FRFT is a critical part of FRFT. The numerical calculation of the FRFT focuses on the following aspects: improving the computational efficiency, approaching the calculation results of the continuous FRFT, and maintaining the properties of the continuous FRFT. Recent decades witness the development of multiple discrete algorithms for the FRFT, e.g. sampling-type DFRFT [1], eigenvector decomposition-type DFRFT [2], linear combination-type DFRFT [3], and so on. Different DFRFT implementations trade off computational efficiency against approximation to the property of continuous FRFT. Here, we introduce two representative implementation ways, matrix multiplication implementation and sampling-type implementation. **In this paper, we select the former one and provide the core code with the supplementary material.**

**Matrix multiplication implementation.** A complete set of eigen-functions of the fractional Fourier transform are the Hermite-Gaussian functions:

$$F^a[\psi_n(x)] = e^{-ian\pi/2}\psi_n(x),$$
$$\psi_n(x) = \frac{2^{1/4}}{\sqrt{2^n n!}}H_n(\sqrt{2\pi}x)\exp\left(-\pi x^2\right) \tag{6}$$

where $H_n(x)$ is the nth-order Hermite polynomial. The spectral expansion of the linear transform kernel is

$$K_a(x, x') = \sum_{n=0}^{\infty} e^{-ian\pi/2}\psi_n(x)\psi_n(x'), \tag{7}$$

The final implementation of DFRFT is:

$$F^a(x, x') = \sum_{n=0, n\neq(N-1+(N)_2)}^{\infty} u_n(x)e^{-ian\pi/2}u_n(x'), \tag{8}$$

where $u_n$ is the nth discrete Hermite-Gaussian function.

**Sampling-type implementation.** The DFRFT of a 1D signal $x(u_0)$ is solved through the following three sequential operations:

1. Chirp multiplication

$$y(u_0) = e^{-j\pi u_0^2 \tan(\alpha/2)}x(u_0). \tag{9}$$

2. Chirp convolution

$$y'(u_\alpha) = A_\alpha \int_{-\infty}^{\infty} e^{j\pi cs(\alpha)(u_\alpha-u_0)^2}y(u_0)\,du_0$$
$$y'(u_\alpha) = A_\alpha \int_{-\infty}^{\infty} h(u_\alpha - u_0)y(u_0)\,du_0 = A_\alpha[h(u_\alpha) * y(u_\alpha)] \tag{10}$$

3. Chirp multiplication

$$X(u_\alpha) = e^{-j\pi u_\alpha^2 \tan(\alpha/2)}y'(u_\alpha). \tag{11}$$

Note that a chirp is a signal in which the frequency varies with time, also called the linear frequency modulation signal. Furthermore, the DFRFT of 2D image signal is implemented by performing 1D DFRFT to each dimension sequentially, which also substantially hinders the speed.

## 3 The relationship between order and performance

In the main manuscript, we choose order=0.5 empirically to represent the fractional domain. In this section, we conduct comprehensive experiments to analyse the relationship between order and

performance. Moreover, we also design the extended version MFRFC++, which learns the fractional order adaptively with the network. Specifically, we choose two representative tasks, object detection with Faster RCNN [4] as backbone and guided image super-resolution with PanNet [5] as backbone. The experimental results are shown in Table 1 and 2. We draw two conclusions from our empirical results. First, different fractional orders in the MFRFC operator all can significantly elevate the performance over the original baseline, with slight difference between different fractional orders. Secondly, the extended version MFRFC++ achieves nearly optimal performance among different fractional orders. Besides, our selection order=0.5 in the main manuscript also gets relatively optimal performance among different fractional orders.

Table 1: The relationship between order and performance on the PASCAL VOC dataset [6] over object detection.

| | Faster RCNN | SFC | MFRFC with different order | | | | | | | | | MFRFC++ |
| | | | 0.1 | 0.2 | 0.3 | 0.4 | 0.5 | 0.6 | 0.7 | 0.8 | 0.9 | Ada |
|---|---|---|---|---|---|---|---|---|---|---|---|---|
| AP50 | 77.30 | 77.90 | 77.80 | 77.80 | 77.80 | 77.60 | 78.00 | 77.90 | 77.70 | 77.70 | 77.50 | 78.00 |
| mAP | 77.26 | 77.89 | 77.77 | 77.78 | 77.79 | 77.61 | 77.96 | 77.94 | 77.66 | 77.69 | 77.53 | 77.98 |

Table 2: The relationship between order and performance on the WorldView II dataset [7] over guided image super-resolution. The performance here is slightly lower due to parallel training on a single GPU. While, the comparison between the data listed below is surely fair under the same setting.

| | PanNet | SFC | MFRFC with different order | | | | | | | | | MFRFC++ |
| | | | 0.1 | 0.2 | 0.3 | 0.4 | 0.5 | 0.6 | 0.7 | 0.8 | 0.9 | Ada |
|---|---|---|---|---|---|---|---|---|---|---|---|---|
| PSNR | 39.552 | 40.514 | 40.667 | 40.560 | 40.548 | 40.556 | 40.611 | 40.563 | 40.561 | 40.660 | 40.414 | 40.603 |
| SSIM | 0.9705 | 0.9739 | 0.9745 | 0.9741 | 0.9743 | 0.9742 | 0.9743 | 0.9740 | 0.9742 | 0.9738 | 0.9741 | 0.9741 |

## 4 Experimental results with single branch

Besides combining different branches to form the SFC or MFRFC operator, we are also interested in the performance of single branch. Specifically, we experiment with three single branches, order=0, order=0.5, and order=1, respectively. We choose two representative tasks, object detection with Faster RCNN as backbone and guided image super-resolution with PanNet as backbone. The experimental results are shown in Table 3 and 4. As shown, single branch performs slightly worse than the integrated operator SFC and MFRFC. For the comparison between different single branches, spectral branch (order=0.5 and order=1) performs better than the spatial branch (which is also the baseline).

Table 3: Experimental results with single branch on the PASCAL VOC dataset [6] over object detection.

| | Order (0) | Order (0.5) | Order (1) | SFC | MFRFC |
|---|---|---|---|---|---|
| AP50 | 77.30 | 77.40 | 76.40 | 77.90 | 78.00 |
| mAP | 77.26 | 77.43 | 76.36 | 77.89 | 77.96 |

Table 4: Experimental results with single branch over guided image super-resolution.

| Methods | WorldView-II | | | | GaoFen2 | | | |
| | PSNR↑ | SSIM↑ | SAM↓ | ERGAS↓ | PSNR↑ | SSIM↑ | SAM↓ | EGAS↓ |
|---|---|---|---|---|---|---|---|---|
| Order (0) | 40.8176 | 0.9626 | 0.0257 | 1.0557 | 43.0659 | 0.9685 | 0.0178 | 0.8577 |
| Order (0.5) | 40.9240 | 0.9638 | 0.0253 | 1.0242 | 43.8439 | 0.9790 | 0.0173 | 0.8009 |
| Order (1) | 41.0039 | 0.9645 | 0.0249 | 1.0013 | 43.8344 | 0.9790 | 0.0173 | 0.7998 |
| SFC | 41.2118 | 0.9663 | 0.0236 | 0.9901 | 44.4965 | 0.9801 | 0.0161 | 0.7761 |
| MFRFC | 41.3428 | 0.9667 | 0.0233 | 0.9898 | 44.6245 | 0.9804 | 0.0146 | 0.6551 |

## 5 The implementaion details

**The architecture of SFC.** In the main manuscript, we introduce the baseline variant Spatial-Frequency Convolution (SFC). Here, we show the architecture design of SFC in Fig. 1.

**Different model implementations.** Our method has three different model settings: the baseline, SFC, and MFRFC. Baseline model performs convolution in the spatial domain. While, the SFC

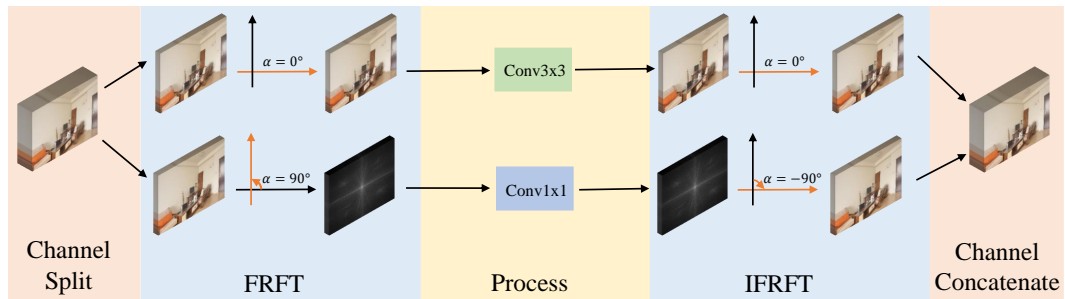

Figure 1: Architecture design of the baseline Spatial-Frequency Convolution (SFC). SFC is comprised of two different order paths: a spatial (p=0) path, and a frequency (p=1) path. IFRFT denotes Inverse FRactional Fourier Transform.

and MFRFC model are implemented by plugging SFC and MFRFC into certain layer of baseline, respectively. Different model implementations are shown in Fig. 2.

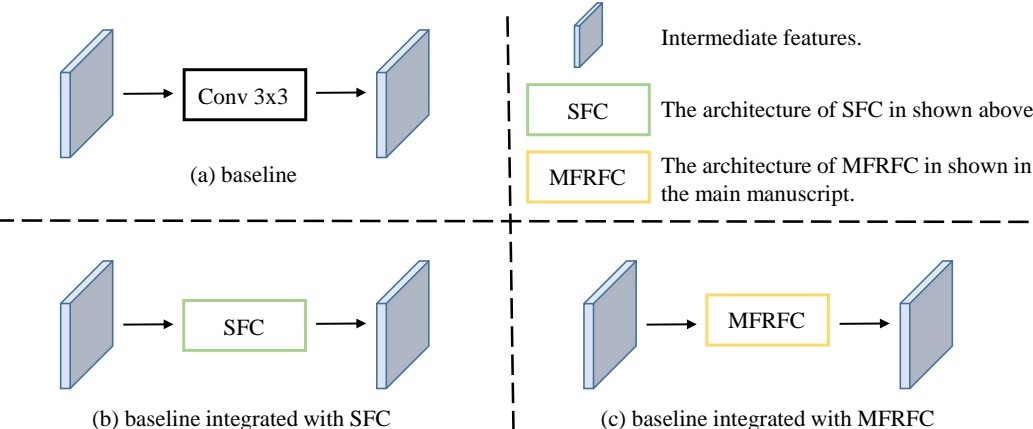

Figure 2: The illustrations of the baseline, the baseline integrated with SFC, and the baseline integrated with MFRFC.

## 6 The periodicity of FRFT

According to Eq. 2, for $p = 4n$ and $p = 4n \pm 2$ where $n \in Z$, the FRFT operator corresponds to the identity operator and parity operator (sequence inversion), respectively. Thus, the operator is periodic at order $p$, with a period value of 4. The periodicity of FRFT can also be interpreted with the spatial-frequency plane. Order $p=4$ equals rotation angle $\alpha = 2\pi$, which represents a cycle on the plane.

$$F^{4n}\{f(x)\} = f(x), F^{4n+1}\{f(x)\} = F(x), F^{4n+2}\{f(x)\} = f(-x), F^{4n+3}\{f(x)\} = F(-x). \quad (12)$$

Further, order range $p \in [0,1]$ is representative in one periodicity as shown in Eq. 12, which theoretically narrows the bound for order selection.

## 7 Energy distribution in FRFT

From the visualization of amplitude and phase spectrums of the FRFT in the main body, characteristics of image energy distribution in the fractional domain are summarized as: it accumulates from the periphery to the center, and the degree of accumulation depends on the degree of order $p$. Specifically, when $p$ is small, most energy is dispersed in the spatial domain. As $p$ gets larger, the energy distribution in the frequency domain shows an apparent upward trend. When $p = 1$, the energy aggregation of the image reaches the strongest.

# 8 Specific example of FRFT in image restoration

In the main manuscript, we demonstrate the merits of FRFT detailly and comprehensively, and design a novel operator fit for various tasks. Besides, some methods introduce the merits of FRFT with task-specific example. For example, [8] shows a specific example to illustrate the advantage of FRFT over traditional Fourier transform on image restoration task. Concretely, given a degraded image, employing optimal filter in the fractional Fourier domain gets much more better results than in the traditional Fourier domain. Need to emphasis that our method applies to various tasks. This task-specific example is for a better understanding of FRFT.

# 9 More qualitative results

In this section, we show more qualitative comparison results over the representative tasks of guided image super-resolution and image dehazing. The qualitative comparison results of guided image super-resolution task on WorldView II dataset are shown in Fig. 3 . We also show more image dehazing results on SOTS dataset in Fig. 4. The obvious contrasts of different methods are annotated in the red box region. Compared with "Original" baseline, our method has finer details for both tasks.

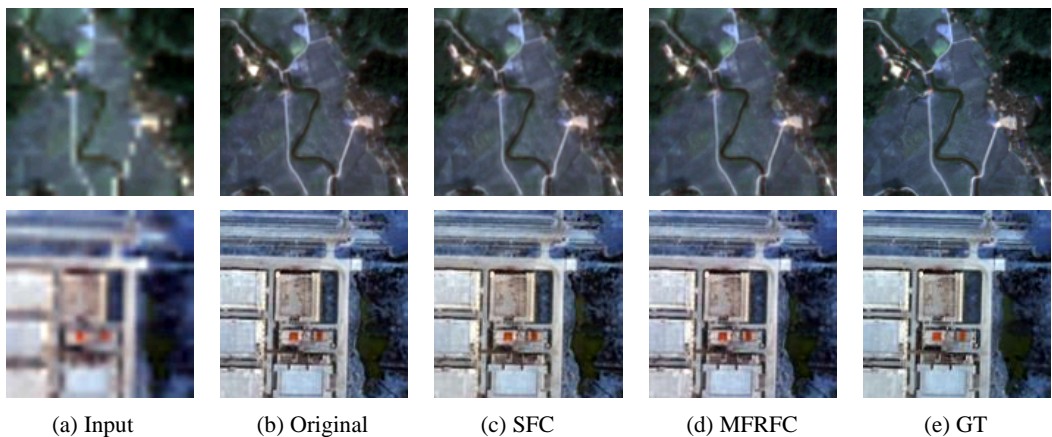

|  (a) Input | (b) Original | (c) SFC | (d) MFRFC | (e) GT |

Figure 3: The visual comparison of guided image super-resolution task over WorldView II dataset.

# 10 Feature map visualization

In this section, we present more visualization results of feature maps, demonstrating the effectiveness of the MFRFC. Fig. 5 and Fig. 6 present the representative example over image dehazing dataset SOTS. As can be seen, the baseline method integrated with our proposed MFRFC has finer details and less artifacts, while the original baseline suffers from detail loss and severe artifacts.

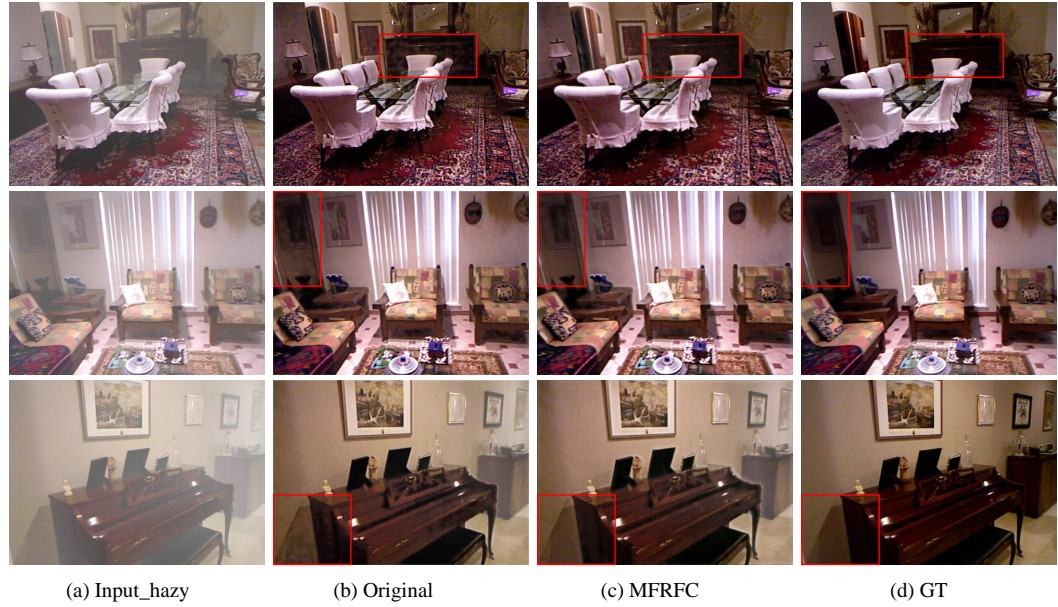

| (a) Input_hazy | (b) Original | (c) MFRFC | (d) GT |

Figure 4: The visual comparison of image dehazing over SOTS dataset.

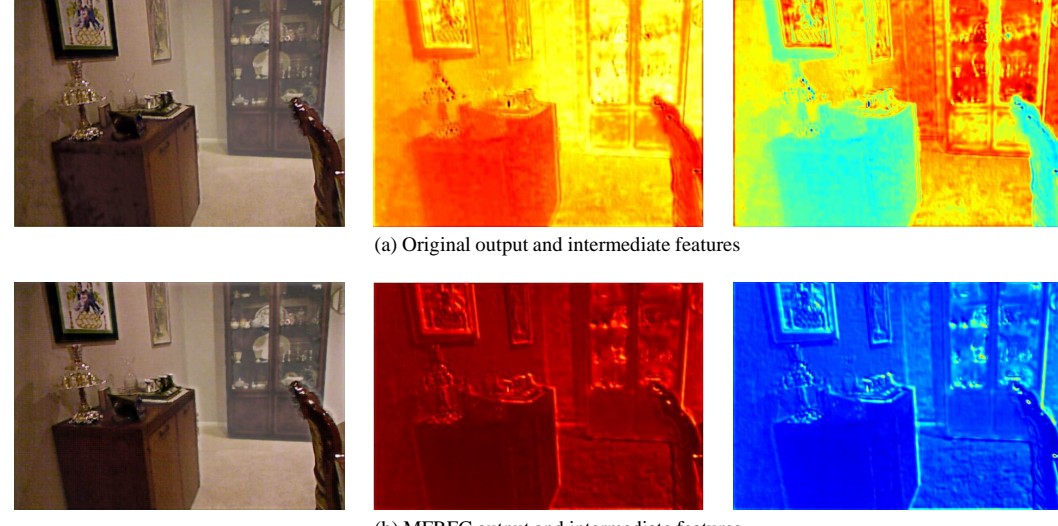

(a) Original output and intermediate features

(b) MFRFC output and intermediate features

Figure 5: The comparison of feature map visualization of different models on image dehazing task.