# OpenReview forum: "Deep Fractional Fourier Transform"
_NeurIPS.cc/2023/Conference — NeurIPS 2023 spotlight_

### Official Review · Reviewer_HR4i · 2023-07-03

**Soundness:** 4 excellent
**Presentation:** 3 good
**Contribution:** 3 good
**Rating:** 8
**Confidence:** 5

**Summary:**

This paper introduces Fractional Fourier Transform to provide comprehensive unified spatial-frequency perspectives for deep learning, and further introduces a basic operator, Multi-order Fractional Fourier Convolution. Besides, this paper experimentally evaluates the effectiveness of MFRFC on various computer vision tasks, including object detection, image classification, guided super-resolution, denoising, dehazing, deraining, and low-light enhancement.

**Strengths:**

There are several strengths here:
1. This paper is of high novelty and originality. Fourier transform is very important and is employed by many deep learning-based methods. But Fractional Fourier transform, a generalized version of Fourier transform, is less explored in deep learning era. This paper introduces FRFT into deep learning, analyses the property of FRFT, and designs a basic convolutional operator.
2. This paper achieves the fast implementation of FRFT, which is very important for the future development of FRFT in the deep learning pipeline.
3.  This paper validates the effectiveness of the MFRFC operator on various computer vision tasks, including high-level tasks (object detection and image classification) and low-level tasks (guided super-resolution, denoising, dehazing, deraining, and low-light enhancement).
4. This paper is clearly written and easy to follow.


**Weaknesses:**

Three are several weakness here:
1. The FRFT itself has a long history and is explored in many research areas. The discussion in the related work part is limited and not comprehensive. For example, the recent work [1] also employs FRFT in image super-resolution.
2. It is necessary to explain the design of MFRFC operator and why such design is optimal. The MFRFC operator employs three paths. But is it possible to apply more paths in the operator. How is the performance and parameters comparison between different number of paths.
3. The MFRFC operator integrates multiple paths/domains, which is superior to single path. But, how is the performance of the operator with only fractional branch compared to other single path. Can single fractional path achieves comparable performance to the full MFRFC operator?

[1] Adaptive Image Super-Resolution Algorithm Based on Fractional Fourier Transform.


**Questions:**

I only have several concerns here:
1. Does there exists a constant optimal fractional order for each task. Besides, the authors exploring the optimal fractional order via learnable parameters. Is there exists other traditional method that determines the optimal fractional?
2. The non-stationary image signals can be better decoupled with FRFT as shown in Figure 1. Can you show some specific examples.


**Limitations:**

The authors adequately addressed the limitations and potential negative societal impact of their work.

---

> ### Author Rebuttal · Authors · 2023-08-07
>
> **1: The research history of FRFT in many research areas.**
> Thanks for the suggestion, we will make a comprehensive discussion in the research history of FRFT in many other research areas in the related work. The reference work you mentioned is quite simple and is not deep learning based, so we did not include it in the submission. We will also add it in the related work.
>
>
> **2: The design choice of MFRFC operator.**
> We devise the MFRFC operator from two main perspectives.
> (1) The number of branches. Our work takes FFC [1] , a spatial and frequency two-branch operator, as baseline. To verify the effectiveness of FRFT, we add one more fractional order branch. The two-branch baseline is designed with the same manner as our three-branch MFRFC operator, with the only difference in fractional branch. Such design is enough to demonstrate the superiority of FRFT over Fourier transform in a fair setting.
> (2) The order of the fractional domain. The order selection of the fractional domain is more important. To this end, we experiment with different fractional order in the supplementary material. Besides, we also conduct experiments with adaptive order. The adaptive order version achieves nearly optimal performance among different fractional orders. Besides, our selection order=0.5 in the main manuscript also gets relatively good performance among different fractional orders.
> [1] Fast Fourier Convolution. NeurIPS, 2020.
>
>
> **3: The performance of the operator with only fractional branch.**
> It is a good question and we explore the performance of the operator with only fractional branch in the supplementary material. We choose two representative tasks, object detection with Faster RCNN as backbone and guided image super-resolution with PanNet as backbone. The experimental results show that single branch performs slightly worse than the integrated operator SFC and MFRFC. For the comparison between different single branches, spectral branch (order=0.5 and order=1) performs slightly better than the spatial branch (which is also the baseline).
>
> **4: About the optimal fractional order.**
> (1) We hold the belief that the optimal fractional order of an image is to do with the task. In other words, it depends on what we want to disentangle from the image. This means that the optimal fractional order is related with the data and task. Thus, for tasks in deep learning paradigm, a fixed optimal fractional order may not exist, since the data is different and uncertain.
> (2) Certain previous method [1] employs entropy maximization to determine the optimal fractional order in hyperspectral anomaly detection task. But this method is coupled with this specific task and data format and this method is not deep learning based. In contrast, setting adaptive order with learnable parameters is a more general way in deep learning paradigm.
> [1] Hyperspectral anomaly detection by fractional Fourier entropy. IEEE Journal of Selected Topics in Applied Earth Observations and Remote Sensing, 2019.
>
>
> **5: Specific examples.**
> It is a good question since specific examples are intuitive for the better understanding of FRFT. We show the decoupling capacity of FRFT for non-stationary signal with a
> previous work [1]. This work explores decoupling degradation from the degraded image with FRFT. FRFT can well decouple degradation and the clean image, filter the degradation, and return the clean image in this case. This well supports our work with specific examples.
> [1] Optimal image restoration with the fractional Fourier transform. JOSA A, 1998.

---

> > ### Comment · Reviewer_HR4i · 2023-08-17
> > **Response to Authors' Comments**
> >
> > Many thanks for your response. After carefully reading authors' rebuttal and other reviewers' comments, my concerns have been addressed. Overall, this paper is with sufficient contributions and convincing experiments，which is suggested to be accepted.

---

### Official Review · Reviewer_6jPA · 2023-07-04

**Soundness:** 2 fair
**Presentation:** 2 fair
**Contribution:** 2 fair
**Rating:** 7
**Confidence:** 5

**Summary:**

+ The paper provides an implementation framework using deep learning for fractional Fourier transform.

**Strengths:**

+ Good deep algo for FRFT.

**Weaknesses:**

- Baselines for different applications are a bit dated.

- The paper could have focused on just one application and dealt deeper.

- The writing is not that good, needs to be precise and with enough motivation.

Addressed.


**Questions:**

- Why there is not fast algo for FRFT?

- Can you tell about the accuracy of Deep FRFT on simulated signals compared to others?

Answered.

---

> ### Author Rebuttal · Authors · 2023-08-07
>
> **1: Baseline methods for different tasks.**
> (1) Our method is not SOTA-oriented. Instead, the key of this paper is that we unlock the FRFT in deep learning paradigm, solving the biggest challenge for the popularization of FRFT: vague characteristics and missing fast implementation. Fourier transform is pretty useful and vastly employed in deep learning era. As a generalized and improved version of Fourier transform, FRFT surely has great application prospect and exploration value.
> (2) We validate the effectiveness of FRFT on various tasks with classical and commonly acknowledged baselines. We select the classical methods as baselines for three reasons: 1. We follow the setting of previous basic operator-based methods [1][2], which also employ classical methods as baselines. 2. The selected baselines are the representative and commonly acknowledged works in the related tasks. In addition, as a general operators, implementing the proposed operator in the standard benchmarks is more fair. 3. The performance of classical methods are highly likely to be reproducible.
> (3) To further solve your misgiving, we also apply our operator to recent SOTA baselines. We choose GPPNN [3] and LACNET [4] for guided image super-resolution task on WorldView-II dataset, NBNet [5] and Restormer [6] for image denoising task on SIDD dataset, URetinex [7] and SNR [8] for low-light image enhancement task on LOL dataset. As can be seen from the following three tables, our method still largely elevates the performance of these SOTA baselines. We will add these experiments and conduct more experiments on remaining tasks in the final version. Besides SOTA baselines, our method may be critical to small networks in the practial deployment,  which shows the irreplaceable and important role of this basic tool and operator.
>
> | Model | Methods |PSNR | SSIM |
> | :-----| ----: | ----: | :----: |
> | GPPNN  | Original | 41.16  | 0.968  |
> |        | SFC      | 40.14  | 0.967  |
> |        | MFRFC    | 41.47  | 0.972  |
> | LACNET | Original | 41.45  | 0.972  |
> |        | SFC      | 41.52  | 0.973  |
> |        | MFRFC    | 41.74  | 0.975  |
>
> | Model | Methods |PSNR | SSIM |
> | :-----| ----: | ----: | :----: |
> | NBNet     | Original | 39.75  | 0.959  |
> |           | SFC      | 39.86  | 0.959  |
> |           | MFRFC    | 39.97  | 0.960  |
> | Restormer | Original | 40.02  | 0.960  |
> |           | SFC      | 40.05  | 0.960  |
> |           | MFRFC    | 40.19  | 0.961  |
>
>
> | Model | Methods | PSNR | SSIM |
> | :-----| ----: | ----: | :----: |
> | URetinex | Original | 21.33  | 0.835  |
> |          | SFC      | 21.82  | 0.837  |
> |          | MFRFC    | 22.35  | 0.839  |
> | SNR      | Original | 24.61  | 0.842  |
> |          | SFC      | 24.84  | 0.846  |
> |          | MFRFC    | 24.98  | 0.848  |
>
>
> [1] Fast Fourier Convolution. NeurIPS, 2020.
> [2] Deep Fourier Up-sampling. NeurIPS, 2022.
> [3] Deep gradient projection networks for pan-sharpening. CVPR, 2021.
> [4] LAGConv: Local-context adaptive convolution kernels with global harmonic bias for pansharpening. AAAI, 2022.
> [5] NBNet: Noise basis learning for image denoising with subspace projection. CVPR, 2021.
> [6] Restormer: Efficient transformer for high-resolution image restoration. CVPR, 2022.
> [7] URetinex-Net: Retinex-based Deep Unfolding Network for Low-light Image Enhancement. CVPR, 2022.
> [8] SNR-Aware Low-light Image Enhancement. CVPR, 2022.
>
>
> **2: The paper could have focused on just one application.**
> (1) Our paper first and comprehensively introduces the characteristics and properties of FRFT in the deep learning paradigm and designs a simple and general operator. Since FRFT is a basic tool and our operator is general, we naturally need to verify the effectiveness and generality of our method on various tasks and classical baselines.
> (2) Our paper unlocks FRFT in deep learning. With this pioneer work, the closer and deeper combination with FRFT and specific tasks is expected to be easier and has great application prospect.
>
>
> **3: The writing quality.**
> Do you have specific suggestions for the writing. Reviewer c5M1 highly praises the writing "The writing is clear and lucid. The experimental settings are clearly described." Reviewer HR4i confirms our writing: "This paper is clearly written and easy to follow."
>
>
> **4: Why there is not fast algo for FRFT?**
> (1) The fast discrete implementation of FRFT is difficult is theory [1][2][3]. Intuitively, the formulation of FRFT shown in equation 1 of the main manuscript is much complicated than that of Fourier transform.
> (2) There exists no fast implementation in practice. Previous discrete implementation of FRFT is much slower than FFT in practical use. There also exists no official package for FRFT.
> [1] The fractional order Fourier transform and its application to quantum mechanics. IMA Journal of Applied Mathematics, 1980.
> [2] Digital computation of the fractional Fourier transform. IEEE Transactions on signal processing, 1996.
> [3] Two dimensional discrete fractional Fourier transform. Signal Processing, 1998.
>
> **5: FRFT on simulated signals.**
> There may be a misunderstanding to our work. Our method validates the effectiveness of FRFT on various tasks. The processed signals are 2-D image signals, instead of simulated signals. What do you mean by simulated signals.

---

> > ### Comment · Reviewer_6jPA · 2023-08-17
> >
> > Satistfied with the rebuttal and other reviews. Raising my rating.

---

> ### Author Response · Authors · 2023-08-14
>
> Hope that we have solved your confusion. We are looking forward to your feedback.

---

### Official Review · Reviewer_c5M1 · 2023-07-04

**Soundness:** 4 excellent
**Presentation:** 3 good
**Contribution:** 4 excellent
**Rating:** 8
**Confidence:** 5

**Summary:**

The paper discusses Fractional Fourier Transform (FRFT) in the context of deep learning based computer vision methods. FRFT is a unified continuous spatial-frequency transform which reflects spatial and frequency representations of images. Based on FRFT, the paper proposes a new convolutional operator (MFRFC) which is more suitable for image signal processing. With MFRFC, the networks achieve consistent performance improvement across several computer vision tasks.

**Strengths:**

+ The paper proposes a good idea to firstly introduce the well-known FRFT in recent deep learning-based methods and achieves substantial performance improvement on various tasks and networks.
+ The proposed method is simple yet effective. The fast implementation of 2D discrete FRFT is vital to the research community. The experimental results show empirical improvement in visual quality in image restoration tasks.
+ The writing is clear and lucid. The experimental settings are clearly described.


**Weaknesses:**

- The concept and properties of FRFT is broad. This paper introduces certain properties of FRFT, but does not comprehensively investigate the FRFT. The proposed operator MFRFC is very effective, while may also be treated as one solution of all the possibilities.
- The paper lacks comparisons with other spectral-related methods that also try to incorporate spectral information in the deep learning pipeline (for example, the FFC method mentioned in the related work).
- MFRFC has three different order paths: a spatial (p=0) path, a spectral (p=1) path, and a fractional order (p=0.5) path. The authors should also evaluate how the order of fractional order path effects performance.
- Visual comparison is incomplete regarding the conducted experiments. The visual results for image denoising and low-light enhancement are missing.


**Questions:**

The introduced FRFT is a spatial-frequency analysis tool of signals. But, there also exists other spatial-frequency analysis tools including wavelet transform and Gabor transform, among which wavelet transform has been widely employed in the deep learning methods. What’s the difference between FRFT and these spatial-frequency analysis tools?

**Limitations:**

The authors have discussed the limitations from different aspects. While they do not include sufficient discussions about the computational cost, which are suggested to be included. Since the authors claim that they solved the fast implementation of FRFT, it is encouraged to clearly compare the speed of the original discrete FRFT, the author’s fast implementation version, and the fast implementation of discrete Fourier transform (FFT).

---

> ### Author Rebuttal · Authors · 2023-08-07
>
> **1: Properties of FRFT.**
> It is true that the properties of FRFT is broad. Here, we introduce the main properties inherent in FRFT. For more properties associated with specific tasks, we believe that with our pioneer work, it will be easier to explore them in the near future. Similar case applies to Fourier transform. Fourier transform itself has only several main properties. However, with the fast implementation FFT, researcher can easily explore the properties of Fourier transform with specific tasks [1][2]. With our work, similar combination can also be made be with FRFT.
> [1] Learning frequency-aware dynamic network for efficient super-resolution. ICCV, 2021.
> [2] Frequency and spatial dual guidance for image dehazing. ECCV, 2022.
>
>
> **2: Comparisons with other spectral-related methods.**
> In fact, we have compared with other spectral-related methods. Previous similar spectral-related methods mainly works in a two-branch manner with spatial and frequency branches. They employ Fourier transform for the frequency branch. In our method, the SFC method works in a similar way with spatial and frequency branches. SFC is designed with the same manner as our three-branch MFRFC operator, with the only difference in fractional branch. Such design is enough to demonstrate the superiority of FRFT over Fourier transform in a fair setting.
>
>
> **3: Relationship between order and performance.**
> We explore the relationship between order and performance in the supplementary material, including different orders and adaptive order. We draw two conclusions from our empirical results. First, different fractional orders in the MFRFC operator all can significantly elevate the performance over the original baseline, with slight difference between different fractional orders. Secondly, the adaptive order version achieves nearly optimal performance among different fractional orders. Besides, our selection order=0.5 in the main manuscript also gets relatively optimal performance among different fractional orders.
>
>
> **4: Difference with other spatial-frequency analysis tools.**
> (1) Gabor and Wavelet transform are also time-frequency analysis tools. The key difference between FRFT and these two spatial-frequency analysis tools is that Gabor and Wavelet transform are indeed special cases of short time Fourier transform. The window function of Gabor is Gaussian function and the window function of Wavelet transform is adaptive. While, FRFT is a generalized version of Fourier transform.
> (2) Gabor has the inherent limitation of short time Fourier transform, such as fixed window function and poor time-frequency resolution balance. Besides, Gabor has more hyper-parameters which makes it hard to optimize. Wavelet transform also suffers from the difficulty of selecting the proper wavelet basis function.
>
>
> **5: Speed of the fast implementation version.**
> We compare the parameters and Flops of MFRFC and vanilla convolution in Table 2 in the main manuscript. As for the speed, the baseline equipped with previous discrete FRFT is about 20 times slower than the original baseline in average. While, the baseline equipped with our fast implementation is about 5\% slower than the original baseline in average. Our method can substantially elevate the performance of baseline method with negligible additional computational burden including parameters, Flops and running speed. Beisdes, we also provide the code for FRFT in the supplement.

---

> > ### Comment · Reviewer_c5M1 · 2023-08-15
> > **Response to Rebuttal**
> >
> > Thanks for the author's thoughtful reply. The rebuttal addressed my concerns well. I was originally positive at the paper. When I checked other reviews and the rebuttal, I decided to raise my rating. Besides, it is better to add these analyses in the rebuttal to the released version.

---

### Official Review · Reviewer_TTF6 · 2023-07-06

**Soundness:** 3 good
**Presentation:** 2 fair
**Contribution:** 3 good
**Rating:** 5
**Confidence:** 5

**Summary:**

This paper proposed a fractional Fourier transform-based module, which can simultaneously exploit the information from spatial and frequency perspectives. With a fast implementation of FRFT, multi-order MFRFC module can be easily incoporated to existing convolutional networks for different tasks.

**Strengths:**

1. A unified spatial-frequency analysis module based on fractional Fourier transofrm is proposed
2. The proposed MFRFC module is applied to several tasks including denoising, deraining, classification, dehazing, detection etc, where the proposed module brings performance gains.

**Weaknesses:**

1. The main issue is the methods for different tasks are not new, e.g., DnCNN has been much inferior to recent denoisers in terms of quantitative metrics. So it should be evaluated whether the proposed module still works for recent denoisers with better performance. I guess the perfmance gains would be not much significant. Currently, I am at the borderline leaning to accept this work. But if my guess is correct, the contribution of this work would be not so significant, and may turn down my rating.

2. It is not clear how the MFRFC module is applied in existing methods for different tasks, e.g., for DnCNN, all the convolutional layers are replaced?


**Questions:**

see weakness

**Limitations:**

see weakness

---

> ### Author Rebuttal · Authors · 2023-08-07
>
> **1: Baseline methods for different tasks.**
> (1) Our method is not SOTA-oriented. Instead, the key of this paper is that we unlock the FRFT in deep learning paradigm, solving the biggest challenge for the popularization of FRFT: vague characteristics and missing fast implementation. Fourier transform is pretty useful and vastly employed in deep learning era. As a generalized and improved version of Fourier transform, FRFT surely has great application prospect and exploration value.
> (2) We validate the effectiveness of FRFT on various tasks with classical and commonly acknowledged baselines. We select the classical methods as baselines for three reasons: 1. We follow the setting of previous basic operator-based methods [1][2], which also employ classical methods as baselines. 2. The selected baselines are the representative and commonly acknowledged works in the related tasks. In addition, as a general operators, implementing the proposed operator in the standard benchmarks is more fair. 3. The performance of classical methods are highly likely to be reproducible.
> (3) To further solve your misgiving, we also apply our operator to recent SOTA baselines. We choose GPPNN [3] and LACNET [4] for guided image super-resolution task on WorldView-II dataset, NBNet [5] and Restormer [6] for image denoising task on SIDD dataset, URetinex [7] and SNR [8] for low-light image enhancement task on LOL dataset. As can be seen from the following three tables, our method still largely elevates the performance of these SOTA baselines.  We will add these experiments and conduct more experiments on remaining tasks in the final version. Besides SOTA baselines, our method may be critical to small networks in the practial deployment,  which shows the irreplaceable and important role of this basic tool and operator.
>
> | Model | Methods |PSNR | SSIM |
> | :-----| ----: | ----: | :----: |
> | GPPNN  | Original | 41.16  | 0.968  |
> |        | SFC      | 40.14  | 0.967  |
> |        | MFRFC    | 41.47  | 0.972  |
> | LACNET | Original | 41.45  | 0.972  |
> |        | SFC      | 41.52  | 0.973  |
> |        | MFRFC    | 41.74  | 0.975  |
>
> | Model | Methods |PSNR | SSIM |
> | :-----| ----: | ----: | :----: |
> | NBNet     | Original | 39.75  | 0.959  |
> |           | SFC      | 39.86  | 0.959  |
> |           | MFRFC    | 39.97  | 0.960  |
> | Restormer | Original | 40.02  | 0.960  |
> |           | SFC      | 40.05  | 0.960  |
> |           | MFRFC    | 40.19  | 0.961  |
>
>
> | Model | Methods | PSNR | SSIM |
> | :-----| ----: | ----: | :----: |
> | URetinex | Original | 21.33  | 0.835  |
> |          | SFC      | 21.82  | 0.837  |
> |          | MFRFC    | 22.35  | 0.839  |
> | SNR      | Original | 24.61  | 0.842  |
> |          | SFC      | 24.84  | 0.846  |
> |          | MFRFC    | 24.98  | 0.848  |
>
>
> [1] Fast Fourier Convolution. NeurIPS, 2020.
> [2] Deep Fourier Up-sampling. NeurIPS, 2022.
> [3] Deep gradient projection networks for pan-sharpening. CVPR, 2021.
> [4] LAGConv: Local-context adaptive convolution kernels with global harmonic bias for pansharpening. AAAI, 2022.
> [5] NBNet: Noise basis learning for image denoising with subspace projection. CVPR, 2021.
> [6] Restormer: Efficient transformer for high-resolution image restoration. CVPR, 2022.
> [7] URetinex-Net: Retinex-based Deep Unfolding Network for Low-light Image Enhancement. CVPR, 2022.
> [8] SNR-Aware Low-light Image Enhancement. CVPR, 2022.
>
>
>
> **2: Implementation of MFRFC.**
> We apply the MFRFC in the middle layer of the network. We find that this is enough to incorporate fractional domain information into the network and can get significant performance improvement with the computational burden introduced by our method negligible.

---

> > ### Comment · Reviewer_TTF6 · 2023-08-17
> >
> > It is necessary to choose baseline models for verification, but I think it is not sufficient to support the effectiveness of proposed method on a wide range of algorithms. Considering the new experiments on SOTA algorithms, I would like to keep my rating for acceptance.

---

### Official Review · Reviewer_Cjk7 · 2023-07-06

**Soundness:** 4 excellent
**Presentation:** 3 good
**Contribution:** 4 excellent
**Rating:** 8
**Confidence:** 4

**Summary:**

1. This technique delves into novel fundamental operators for deep learning, the Fractional Fourier Transform (FRFT), exploring a new perspective of signal processing between two orthogonal domains (spatial and frequency domains).

2. This technique have implemented a fast and differentiable Multi-order Fractional Fourier Transform Convolution, an elegant combination of deep learning and traditional image processing.

3. Unignorably, as an theoretical extension of FFC, it is expected to make positive contributions to the research community.

**Strengths:**

This work provides a solid foundation for delving into this unexplored territory for deep learning. A unified analysis tool in the spatial-frequency domain, known as the Fractional Fourier Transform (FRFT), has been introduced, accompanied by sufficient theoretical analysis with following points:

+ The spatial and frequency domains have been extensively explored, while the intermediate chaotic region between the two has been underestimated.

+ The author has developed a fast implementation of the 2D FRFT, enabling comprehensive image processing from multiple perspectives in the spatial-frequency plane.

+ Sufficient experimentation. The operators were experimentally evaluated on a range of vision tasks and results demonstrate its substantial performance improvements.

+ Overall, the study is solid, authors also provide executable code implementation. It is a valuable supplement to deep learning-based vision toolkits.

**Weaknesses:**

There are also few concerns:
- The three-branch design in MFRFT indeed reasonable. However, it is worth considering whether there is a justifiable explanation for utilizing 1x1 convolutions to process signals in fractional domains.
- Additionally, has the author taken into account the possibility of handling this chaotic information through the MLP layers instead of a unified convolutional operator?

-The introduced FRFT is mainly applied to CNN-based architectures. The transformer has also demonstrated strong capacities and performance. Thus, the  authors may also need to discuss the possibility of introducing FRFT in transformer architecture. The basic operator for transform may be different from that for CNN.
- How FRFT in low-level vision tasks and high-level vision tasks. In FFC, it only conduct experiments in high-level vision tasks. Why authors choose low-level vision tasks for experiments conduction?
- Several minor issues that need to be modified.
(1) Inconsistent abbreviation for the citation of figures. Figure in line 248 and Fig. in line 249.
(2) Figure 1 is easy to understand, but the detail of this figure may need further improving for beauty.
(3). The caption for Table 2 is abnormally long, compared to other Figures and Tables.
(4) The evaluation for guided image super-resolution task adopts two more metrics than all other evaluated low-level tasks, which may be redundant.

**Questions:**

In Figure 2, compared to the spatial and frequency domains, the fractional amplitude spectra exhibit a scaling effect. Why is this phenomenon present?

**Limitations:**

Yes. The limitations and potential negative societal impact of this paper have been addressed in this paper.

---

> ### Author Rebuttal · Authors · 2023-08-07
>
> **1: 1x1 convolutions in fractional domains.**
> FRFT is a generalized and extended version of Fourier transform. For Fourier transform, spectral theory demonstrates the existence of operator duality between convolution in the spatial domain and element-wise multiplication in the spectral domain, and thus 1x1 convolution is a default setting in spectral domain [1] [2]. Correspondingly, we also employ 1x1 convolution for fractional fourier domain.
> [1] Fast Fourier convolution. NeurIPS, 2020.
> [2] Deep Fourier Up-sampling. NeurIPS, 2022.
>
> **2: FRFT in CNN, MLP and Transformer architectures.**
> (1) Our work is based on FFC [1] with an theoretical extension. Following this baseline work, we design a convolutional operator for CNN-based architectures. Most of previous networks are CNN-based, endowing our method with promising application prospect.
> (2) Fourier transform has been explored in both MLP [2] and Transformer [3] architectures. Our work is an improved version of Fourier transform, and can also been explored in these architectures as a future work.  For example, GFNet [3] replaces the self-attention sub-layer with the frequency filter layer via Fourier transform. Such operation can be directly replaced with FRFT. Besides, FRFT can be applied to more architectures and tasks since our work unlocks FRFT in deep learning.
> [1] Fast Fourier convolution. NeurIPS, 2020.
> [2] Fourier features let networks learn high frequency functions in low dimensional domains. NeurIPS, 2020.
> [3] Global filter networks for image classification. NeurIPS, 2021.
>
>
> **3: FRFT in both low-level and high-level vision tasks.**
> (1) FRFT is a basic tool and our designed MFRFC is a general operator. Thus we conduct comprehensive experiments on both low-level and high-level vision tasks to validate the effectiveness of our method.
> (2) Besides the consideration of comprehensive verification on different tasks, we find support that FRFT is closely related to low-level vision tasks [1]. This also motivates us to verify our method on low-level vision tasks.
> [1] Optimal image restoration with the fractional Fourier transform. JOSA A, 1998.
>
>
> **4: Scaling effect in the fractional amplitude spectra.**
> The scaling effect in the fractional amplitude spectra is the inherent characteristic of FRFT. It is to do with the projection of fractional domain signal on the spatial domain, as shown in Figure 1 in the main manuscript. As the fractional order varies from 0 to 1, the energy projection on the spatial domain gets less, demonstrating as the scaling effect. This phenomenon is also explained as the energy distribution property of FRFT in the supplementary material.

---

> > ### Comment · Reviewer_Cjk7 · 2023-08-19
> >
> > Thanks for your response, all of my concerns have been well-addressed. Therefore, I decide to raise my score.

---

### Decision · Program_Chairs · 2023-09-21

**Decision:**

Accept (spotlight)

**Comment:**

All reviewers recommended to accept this paper, with 3 "strong accepts".
Connecting the bridge between deep learning and Fractional Fourier Transform was very well received by the reviewers, and the work can serve as a foundation for future improvement in deep learning.
The AC agrees with the reviewers, and recommend to accept this paper.